# Partial Discharge Detection Technology for Switchgear Based on Near-Field Detection

Chunguang Suo [1], Jingjing Zhao [1], Xuehua Wu [1,*], Zhipeng Xu [2], Wenbin Zhang [2] and Mingxing He [1]

1    College of Science, Kunming University of Science and Technology, Kunming 650504, China
2    College of Mechanical and Electrical Engineering, Kunming University of Science and Technology, Kunming 650504, China
*    Correspondence: xhwu@kust.edu.cn

**Abstract:** In view of the fact that the partial discharge (PD) signal energy is mainly concentrated below hundreds of megahertz, the ultra-high frequency part of the energy is weak, and the interior space of the switchgear is narrow, this paper proposes a new method for PD detection of the switchgear based on near-field detection. Firstly, based on the principle of PD, the field characteristics of the signal in the switchgear are analyzed. After that, the probe is designed with an electric small loop structure. Based on its equivalent circuit, its measurement principle and amplitude frequency characteristics are analyzed. The influence of probe size and material on amplitude frequency characteristics is obtained by using simulation software High Frequency Structure Simulator (HFSS), and the probe parameters suitable for PD detection in the switchgear are determined. Finally, the performance of the probe is measured by network analyzer, and the PD signal is tested on the simulated PD test platform. The results show that the probe works in the frequency band of 10–200 MHz and can receive PD signals containing more energy information. In the operating frequency band, the reflection coefficient of the probe port is very large, and its interference to the signal near field is particularly small. The probe also has good frequency response characteristics, and the fluctuation in the frequency band is less than 5 dB, which can obtain more accurate PD signal characteristics in subsequent processing. In addition, the probe is passive, with dimensions of 166 mm in length, 104 mm in width, and 2 mm in thickness, which is suitable for placing in the switchgear with small internal space. The results of PD receiving test show that the probe can reflect the occurrence of PD remarkably and accurately.

**Keywords:** partial discharge detection; near-field detection; switchgear; magnetic field probe





## 1. Introduction

Switchgear is an important electrical equipment that is widely used in power systems and directly supplies power to distribution network and users. When its failure causes power failure, it brings serious economic losses and social losses. Most of the faults in the switchgear are caused by internal insulation faults [1]. Insulation defects are often accompanied by partial discharge (PD). Therefore, carrying out PD detection on the switchgear can effectively evaluate the insulation status of the equipment, find its latent discharge fault, improve the test and maintenance efficiency of the switchgear, and ensure the safe and reliable operation of the equipment [2].

When PD occurs in the insulation medium, many electrical (such as electric pulse, increase of dielectric loss, and electromagnetic wave emission) and non-electrical (such as light, heat, noise, chemical change, and change of gas pressure) phenomena will occur. Therefore, the detection methods can be roughly divided into electrical detection method and non-electrical detection method. Generally, the non-electrical detection method has low sensitivity. The electrical detection method mainly includes pulse current method [3] and ultra-high frequency (UHF) detection method [4]. The pulse current method measures the pulse current caused by PD at the coupling capacitor side or from the neutral point

or grounding point of power equipment through the Rogowski coil by measuring the impedance, so as to obtain the discharge information such as the apparent discharge quantity and discharge phase. It is also the only method with international standards IEC 60270:2000 and GB/T 7354-2003 that can quantify PD in PD detection. However, it is generally contact measurement, and the lower frequency part of the measured PD signal spectrum is generally from several kHz to tens of MHz [5,6]. The signal contains little information, which is commonly used in the type test, factory test, and other offline tests of electrical equipment such as switchgear. UHF detection method is a non-contact measurement of UHF 300–3000 MHz far-field electromagnetic wave signal radiated by PD through UHF antenna. It has the advantage of avoiding field interference below 300 MHz and has strong anti-interference performance. It has become the most popular and widely used online monitoring method for PD of switchgear at present [7–12]. However, it also avoids PD signal with rich information frequency band [13]. The UHF part of the signal has very weak energy, and complex refraction and reflection will occur when it propagates in the switchgear under the influence of the environment, with a large amplitude attenuation [14,15]. It is difficult to carry out quantitative and pattern recognition of PD. Although many substations have established perfect condition monitoring systems based on UHF detection technology, insulation failures continue to occur [16,17], which means that many PD have not been monitored. This is because [18,19] some submillimeter cracks may also appear in the insulator, and the discharge caused by it is mainly glow discharge. The discharge frequency is basically below 150 MHz, which cannot reach the UHF frequency band. In addition, as a common type of PD, tip discharge signals are mainly distributed below 200 MHz [20,21].

Based on the above analysis, combined with the actual space in the switchgear, this paper proposes a new method of PD detection of switchgear based on near-field detection. This method uses magnetic field probe to couple magnetic field generated by PD signal, which can realize non-contact online monitoring of PD. The probe is small in size, and its working frequency band is below 200 MHz. It can be placed in the switchgear to obtain signals in the energy concentrated frequency band, which is expected to achieve the goal of quantifying the PD signal, estimating the PD state of the switchgear and diagnosing the insulation.

Firstly, based on the generation principle of PD pulse current, combined with the actual environment and size of the switchgear, this paper analyzes the field characteristics of pulse current generated in the switchgear, and verifies the rationality and necessity of PD near-field detection. Then, combined with the application environment, it is proposed that the probe adopts a plane electric small loop structure, and its measurement principle and amplitude frequency characteristics are analyzed according to its equivalent circuit. On this basis, how to broaden the working frequency band of the probe on the basis of miniaturization of the probe is discussed, and the corresponding solutions are proposed. The influence of parameters such as probe size on its frequency characteristics is analyzed through simulation, and the corresponding parameters of the probe are finally determined. Finally, the network analyzer is used to measure the performance parameters of the probe, and the probe receiving PD signal is tested on the simulated PD platform built in the laboratory to verify the performance of the probe.

## 2. Principle of Near-Field Detection of PD in Switchgear

### 2.1. PD Principle of Switchgear

This paper takes Schneider SM6 medium voltage switchgear as an example, and its basic internal environment is shown in Figure 1. It can be seen that the internal space of the switchgear is narrow, the structure is complex, the components are various, and the insulation pressure is large. Therefore, it is more prone to insulation failure than other power equipment, which brings huge hidden trouble to the safe operation of equipment. There are often some weaknesses in the insulation of switchgear equipment, such as air gaps or bubbles in some casting, extrusion, or layer wound insulation. The breakdown field

strength and dielectric constant of air are smaller than those of solid medium, so under the action of applied voltage, these air gaps and bubbles will discharge first. In addition, there are some edges and burrs, metal protrusions, and metal tips in the busbar and components in the switchgear [22], which are also easy to discharge.

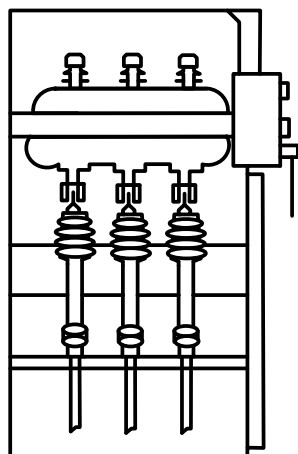

**Figure 1.** Internal environment of the switchgear.

The principle of PD is often explained by the three-capacitance model [23], as shown in Figure 2, where $C_g$ represents the capacitance of the air gap or bubble, $C_{b1}$ and $C_{b2}$ represent the capacitance of the medium in series with $C_g$, respectively, and $C_a$ is the capacitance of the remaining part of the medium. Adding an AC voltage $u_t$ between the electrodes:

$$u_t = U_{max} \sin \omega t \tag{1}$$

where $U_{max}$ is the maximum value of the AC voltage applied across the electrodes and $\omega$ is the angular frequency.

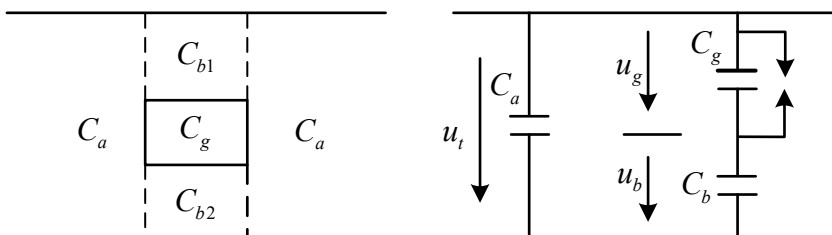

**Figure 2.** Three-capacitance model for PD.

Then, the voltage $u_g$ appearing across the air gap or bubble $C_g$ is:

$$u_g = \frac{C_b}{C_g + C_b} U_{max} \sin \omega t \tag{2}$$

As the air gap is very small, $C_g$ is much larger than $C_b$, so $u_g$ is much smaller than $u_t$. The applied voltage, the voltage in the air gap, and the current change during PD are shown in Figure 3. The applied voltage $u_t$ rises and the voltage $u_g$ in the air gap rises with it, when $u_t$ rises to the starting discharge voltage $U_s$, i.e., $u_g$ reaches the discharge voltage $U_g$ of $C_g$, $C_g$ discharges and the voltage on it quickly drops from $U_g$ to $U_r$, completing a discharge. Afterwards, as the applied voltage $u_t$ rises, the voltage $u_g$ on the air gap also continues to rise, when it rises again to $U_g$, $C_g$ discharges again, the discharge goes out again, the voltage on the air gap drops to $C_r$ again, and so on, the voltage on $C_g$ changes between $U_g$ and $U_r$, at this time, there is a pulse current generation on the external circuit through $C_g$.

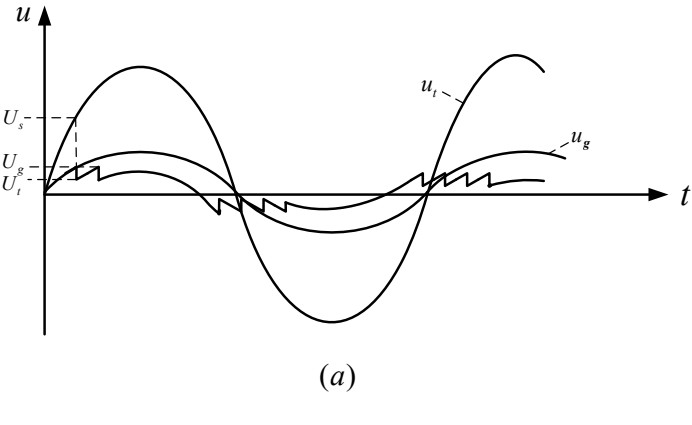

(*a*)

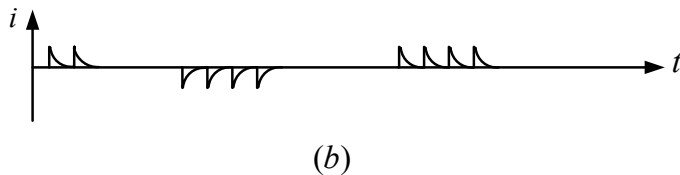

(*b*)

**Figure 3.** Changes in the air gap during PD. (**a**) Voltage; (**b**) current.

*2.2. Near-Field Characteristics of PD Pulse Currents in Switchgear*

When a PD occurs in the switchgear, it is also accompanied by a pulse current, which according to Maxwell's first equation:

$$\nabla \times \vec{H} = \vec{J} + \frac{\partial \vec{D}}{\partial t} \tag{3}$$

where $\vec{H}$ is the magnetic field strength, $\vec{J}$ is the conduction current density, and $\vec{D}$ is the electric displacement vector. According to Equation (3), conducting current and time-varying electric field can generate magnetic field, so that time-varying magnetic field will be generated in the switchgear.

To analyze the field characteristics generated by the pulse current in the switchgear, the current element can be used for equivalent analysis, and the magnetic field distribution characteristics generated by the current element can be solved according to the wave equation. In order to simplify the analysis process of this problem, the vector potential $\vec{A}$ and the scalar potential $\varphi$ can be introduced. Combining the definition of vector potential $\vec{A}$ and scalar potential $\varphi$, vector identity and Lorentz condition, the wave equation satisfied by the vector potential $\vec{A}$ can be obtained:

$$\nabla^2 \vec{A} - \mu\varepsilon \frac{\partial^2 \vec{A}}{\partial t^2} = -\mu \vec{J} \tag{4}$$

where $\mu$ is the magnetic permeability and $\varepsilon$ is the dielectric constant. From Equation (4), the relationship between the vector potential $\vec{A}$ and the current source $\vec{J}$ is known. Combining the solution method of the wave equation [24], i.e., first obtain the general solution of the equation, and then combine the conditions of the definite solution to obtain the electromagnetic vector potential $\vec{A}$ of the current element:

$$\vec{A} = \int_v \vec{J} \frac{\mu e^{-jkr}}{4\pi r} dv \tag{5}$$

In Equation (5), $k = \omega\sqrt{\mu\varepsilon}$ is the wave number. As can be seen in Figure 4, the length of the element is $l$, the vector potential of the current element is:

$$\vec{A} = \vec{e}_z \frac{\mu_0 I l e^{-jkr}}{4\pi r} \tag{6}$$

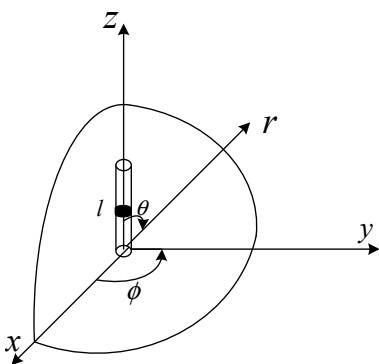

**Figure 4.** Schematic of pulse current.

The conversion gives the expression for the vector potential $\vec{A}$ in the spherical coordinate system as:

$$\vec{A} = \vec{e}_r \frac{\mu_0 I l}{4\pi} \cdot \frac{e^{-jkr}}{r} \cos\theta - \vec{e}_\theta \frac{\mu_0 I l}{4\pi} \cdot \frac{e^{-jkr}}{r} \sin\theta \tag{7}$$

Combined with the definition of the vector potential $\vec{A}$ and the constitutive relations of linear and isotropic media, The expression of magnetic field intensity generated by pulse current can be obtained as follows:

$$\vec{H} = \vec{e}_\phi j \frac{I l}{2\lambda r} \sin\theta (1 + \frac{1}{jkr}) e^{-jkr} \tag{8}$$

From Equation (8), the magnetic field generated by placing the pulsed current on the $z$-axis has only the $\phi$-directional component of the magnetic field strength. In general, the region of $r \ll \lambda$ is called the near field [25]. Combined with the analysis in the previous section, the PD signal energy is mainly concentrated below 200 MHz, and its wavelengths $\lambda$ are all greater than 1.5 m. For some common models [26]: SM6, GG-1A (F), XGN2-12, JYN6-12, KYN1-10 switchgear, the cabinet width and depth are generally below 1 m, and the width of SM6 model switchgear can be up to 0.5 m, combined with the actual environment inside the switchgear. Figure 1 shows that the discharge position reaches the position where the built-in sensor of the PD is placed at a distance $r$ far less than 1.5 m. That is, for the signals in these energy-concentrated frequency bands, the entire switchgear is in the near-field region of these signals. Neglecting secondary factors, the strength of the magnetic field generate by the near field of the pulsed current is:

$$\vec{H} = \vec{e}_\phi \frac{I l \sin\theta}{4\pi r^2} \tag{9}$$

From Equation (9), it can be seen that in the near-field region, the magnetic field generated by the pulsed current in the switchgear is the same as the constant current element expression calculated by Biot–Savart Law in the static magnetic field. The time-varying electric field generated by the time-varying magnetic field can be obtained by calculation:

$$\vec{E} = -\vec{e}_r j \frac{I l}{2\pi\omega\varepsilon} \cdot \frac{\cos\theta}{r^3} - \vec{e}_\theta j \frac{I l}{4\pi\omega\varepsilon} \cdot \frac{\sin\theta}{r^3} \tag{10}$$

Therefore, in the near-field region, the average energy flow density vector $\vec{S}_{av}$ of the electromagnetic field generated by the pulsed current is:

$$\vec{S}_{av} = \frac{1}{2}\text{Re}[\vec{E} \times \vec{H}^*] = 0 \tag{11}$$

As can be seen from Equation (11), if the result caused by minor factors in the field representation is ignored, it can be regarded as there being no electromagnetic power output in the near field area of the pulse current, that is, the energy is bound around the pulse current. Therefore, in order to obtain most of the energy information of the PD signal, the near-field detection should be studied. It is expected to realize the goal of on-line detection of PD, quantification of PD signal, estimation of PD state of switchgear, and insulation diagnosis.

## 3. Magnetic Field Probe Design

### 3.1. Design Requirements

The magnetic field probe designed in this paper will be placed in the switchgear to detect the PD signal. In view of the characteristics of the detected signal and the application environment, and in order to be able to quantify and identify the pattern of the local discharge signal in the subsequent study, the following design requirements are put forward for the magnetic field probe:

1.  Combined with the above analysis of the PD signal in the switchgear, the operating frequency band of the probe is set at 10–200 MHz;
2.  Since the magnetic field probe designed in this paper is a near-field receiving antenna, its role is to couple the electromagnetic energy from the point to be measured to output, but does not have electromagnetic energy input from the probe because the signal from the port into the probe will be radiated out through the probe. Since it is a relatively weak PD signal, the probe of this radiation is a kind of interference; this interference should be as small as possible, so the reflection coefficient in the probe input is as large as possible;
3.  In order to get the accurate characteristics of the PD signal in the subsequent data analysis, the probe is required to have a smooth response to the measured magnetic field in the working band, that is, for signals of different frequencies, theoretically, as long as the field strength amplitude is equal, the data output by the probe should be equal, but the actual design process is not able to achieve absolute equality, so it is required to be as smooth as possible, and hopefully its frequency characteristics fluctuate less than 5 dB [27] as far as possible, which is conducive to the subsequent processing of the signal.
4.  There are many components in the switchgear, and the structure is complex. The size of the built-in sensor is too large, which has a great impact on the electrical environment in the switchgear, and will introduce new insulation defects. This requires the magnetic field probe to be as small as possible without abandoning the above performance requirements.

### 3.2. Working Principle

From the above design requirements, the probe needs to detect the PD signal wavelength range of 1.5–30 m. In the common resonant antenna design probe, the probe size is generally $\frac{\lambda}{2}$ or $\frac{\lambda}{4}$, the size is very large, it is difficult to be placed in the switchgear, And the size will be too large for the electrical environment. In the switchgear has a greater impact, so this paper uses the electric small loop structure to design the magnetic field probe. In general, the maximum geometry of the antenna is much smaller than the working

wavelength of the antenna, i.e., satisfying Equation (12) can be called an electric small loop antenna, which is defined by H.A. Wheeler [28] as follows:

$$\frac{l}{\lambda} \leq \frac{1}{2\pi} \tag{12}$$

where $l$ is the maximum geometric size of the antenna, $\lambda$ is the working wavelength.

The schematic diagram of the operation of the magnetic field probe is shown in Figure 5, and its basic principle is Faraday's law of electromagnetic induction:

$$V_m = -\oint_s \frac{\partial \vec{B}}{\partial t} \cdot d\vec{S} \tag{13}$$

where $\vec{B}$ is the magnetic induction intensity, $S$ is the area through which the magnetic induction lines pass, and $V_m$ indicates the magnitude of the induced electric potential.

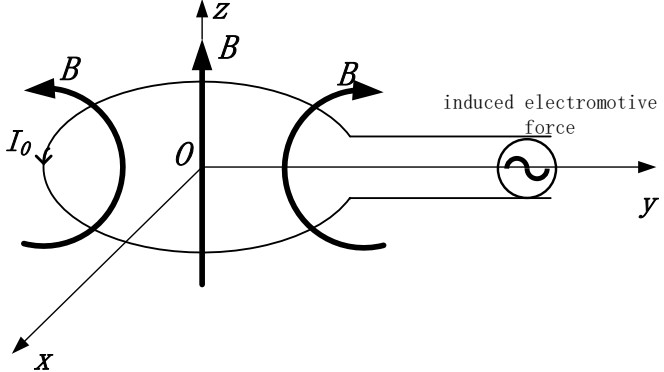

**Figure 5.** Schematic diagram of magnetic field probe.

For the magnetic field probe, the total length of the wire wound around the loop is much smaller than its operating wavelength, so it can be assumed that the magnetic field through the loop area $A$ is uniformly distributed, which further gives:

$$V_m = -\mu A \frac{\partial H}{\partial t} \tag{14}$$

where $\mu$ is the magnetic permeability of the medium where there are magnetic induction lines passing through the electric small loop antenna.

When the size of the probe is small compared to the operating wavelength, it is essentially an inductor with a small amount of radiation, a capacitor, or some combination of both [29]. Therefore, the magnetic field probe can be analyzed using the lumped parameter theory, and its Thevenin's equivalent circuit [30] is shown in Figure 6, where $V_m$ is the probe induced voltage, $R_0$ is the internal resistance, $L_0$ is the self-inductance, $C_0$ is the stray capacitance, and $U_0$ is the output voltage at the load side.

Based on the above circuit, combined with Kirchhoff's voltage law (KVL) and Kirchhoff's current law (KCL), the equation can be obtained:

$$V_m = L_0 C_0 \frac{\partial^2 U_0}{\partial t^2} + \left( R_0 C_0 + \frac{L_0}{R_L} \right) \frac{\partial U_0}{\partial t} + \left( \frac{R_0}{R_L} + 1 \right) U_0 \tag{15}$$

Under zero initial conditions, the combined Equation (14) and the Equation (15) after Laplace transformation can obtain:

$$-\mu A p H(p) = \left[ L_0 C_0 p^2 + \left( R_0 C_0 + \frac{L_0}{R_L} \right) p + \left( 1 + \frac{R_0}{R_L} \right) \right] U_0(p) \tag{16}$$

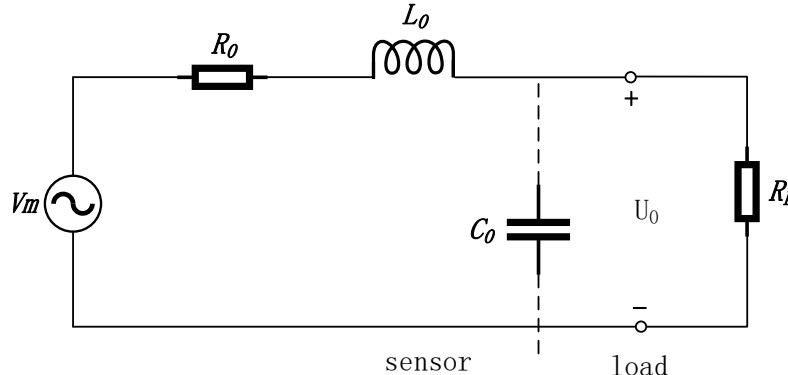

**Figure 6.** Thevenin's equivalent circuit of magnetic field probe.

This leads to the transfer function of the probe:

$$T(p) = \frac{U_0(p)}{H(p)} = \frac{-\mu A p}{\left[L_0 C_0 p^2 + \left(R_0 C_0 + \frac{L_0}{R_L}\right)p + \left(1 + \frac{R_0}{R_L}\right)\right]}$$
$$= \frac{-\mu A p}{L_0 C_0 (p - p_1)(p - p_2)} \tag{17}$$

where $p$ is a complex variable and $p_{1,2} = -\delta \pm \sqrt{\delta^2 - {\omega'}_0^2}$, where $\delta = \frac{R_0}{2L_0} + \frac{1}{2R_L C_0}$, $\omega'_0 = \frac{1}{\sqrt{L_0 C_0}}\sqrt{\left(1 + \frac{R_0}{R_L}\right)} = \sqrt{p_1 \cdot p_2}$, when $\delta > \omega'_0$, $p_1$ and $p_2$ are two different real roots.

The frequency characteristic of the system is a certain steady-state characteristic of the system under the excitation of the sinusoidal signal [31], as long as the $p$ in the system function is replaced by $j\omega$, the expression of the system frequency response can be obtained:

$$T(j\omega) = |T(j\omega)|^{j\varphi(\omega)} = -\frac{\mu A \cdot R_L}{R_L + R_0} \cdot \frac{j\omega}{\left(\frac{j\omega}{p_1} - 1\right)\left(\frac{j\omega}{p_2} - 1\right)} \tag{18}$$

$$|T(j\omega)| = \frac{\mu A \cdot R_L}{R_L + R_0} \cdot \frac{\omega}{\sqrt{1 + \left(\frac{\omega}{\delta - \sqrt{\delta^2 - {\omega'}_0^2}}\right)^2}\sqrt{1 + \left(\frac{\omega}{\delta + \sqrt{\delta^2 - {\omega'}_0^2}}\right)^2}} \tag{19}$$

where $\omega$ is the angular frequency and $\varphi(\omega)$ is the phase angle. The general representation of the logarithmic amplitude frequency characteristics of the system function can be obtained from Equation (19):

$$G(\omega) = 20\log|T(j\omega)|(dB) = 20\log\frac{\mu A \cdot R_L}{R_L + R_0} + 20\log\omega$$
$$-10\log\left[1 + \left(\frac{\omega}{\omega_L}\right)^2\right] - 10\log\left[1 + \left(\frac{\omega}{\omega_H}\right)^2\right](dB) \tag{20}$$

In Formula (20), $\omega_L$, $\omega_H$ are respectively the upper and lower cut-off angular frequency points of the probe, and the probe operates between these two cut-off frequencies, the bandwidth is $\Delta\omega = \omega_H - \omega_L = 2\sqrt{\delta^2 - {\omega'}_0^2}$. The larger the $\delta$, the wider the region.

When $\delta \gg \omega'_0$, the joint equivalence infinitesimal $(1 + x)^a - 1 \sim ax$, the lower cutoff frequency:

$$\omega_L = \delta\left[1 - \sqrt{1 - \left(\frac{\omega_0}{\delta}\right)^2}\right] \approx \frac{R_L + R_0}{L_0} \tag{21}$$

$$f_L \approx \frac{R_L + R_0}{2\pi L_0} \tag{22}$$

Also at $\delta >> \omega'_0$, the upper cutoff frequency of the measuring circuit is:

$$\omega_H = \delta \left[ 1 + \sqrt{1 - \left(\frac{\omega'_0}{\delta}\right)^2} \right] \approx \frac{1}{R_L C_0} \tag{23}$$

$$f_H \approx \frac{1}{2\pi R_L C_0} \tag{24}$$

From Equation (24), it is known that the upper cut-off frequency of the probe is mainly affected by the stray capacitance of the probe, i.e., the equivalent size of the electric small loop, i.e., for a single-turn electric small loop probe, there is the following relationship [32] between the frequency $f_{max}$ of the highest measurable signal and its loop radius $b$:

$$f_{max} = \frac{0.2c}{2\pi b} \tag{25}$$

where, $c$ is the speed of light in vacuum.

### 3.3. Design and Simulation

In order to facilitate the placement of the probe in the switchgear, the overall structure of the probe adopts an electric small loop structure fed by coplanar waveguide, as shown in Figure 7, where $a$ is the line radius, $b$ is the loop radius. the probe is mainly composed of two parts: the signal line and the ground plane, the signal line and the ground plane are located in the same plane, the signal line is located in the middle, the ground plane is located on the left and right sides. Based on the principle of electric small loop, the signal line of the magnetic field probe is wound into a loop structure, which is used to couple the magnetic field energy. In addition, the end of the signal line that is wound into a loop structure is shorted to the ground plane, and the other end forms a coplanar waveguide transmission line structure with the ground plane, thus forming a closed loop.

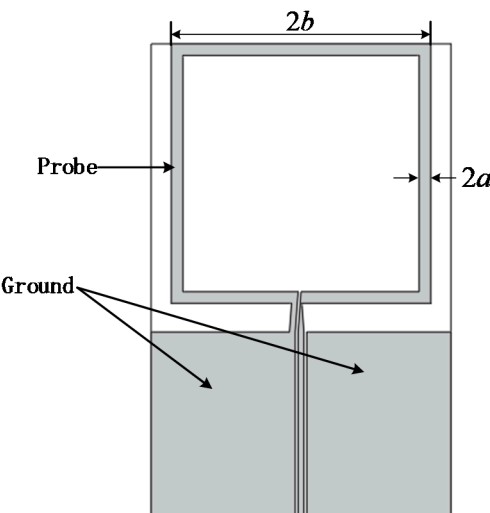

**Figure 7.** Schematic diagram of magnetic field probe structure.

Combined with the design requirements of the magnetic field probe, this section focuses on establishing the probe model in High Frequency Structure Simulator (HFSS) simulation software, obtaining the frequency characteristics of the probe port, analyzing the law, and thus determining the probe parameters that meet the design requirements of this paper.

### 3.3.1. Effect of Line Radius on Frequency Characteristics

The magnetic field probe is modeled on HFSS according to Figure 7. The initial radius of the probe is set to 33 mm, and the substrate material is epoxy resin with relative permittivity $\varepsilon_r$ of 4.4. The material properties of the probe and the ideal and radiation boundary conditions were set up and a piece to be tested was established. The distance between the piece to be tested and the probe was 0.1 m. the input port of the object to be tested was set as 2 ports and wave excitation added, then a pulse current was generated on the object to be tested, and magnetic field was generated in its surrounding space. The probe torus was perpendicular to the magnetic field direction, and the output port of the probe was set as port 1. $S_{12}$ is the transmission parameter from port 2 to port 1:

$$S_{12} = \left. \frac{b_1}{a_2} \right|_{a_1=0} \tag{26}$$

where $b_1$ represents the normalized outgoing wave voltage at port 1, $a_1$, $a_2$ represent the normalized incoming wave voltage at ports 1 and 2, respectively. $S_{12}$ can be used to represent the frequency response characteristics of the probe. Within the 300 MHz frequency range, the magnetic field probes with line radius of 2 mm, 7 mm, and 12 mm were simulated, and then the frequency response characteristics of the probe under the three line radius sizes were obtained as shown in Figure 8.

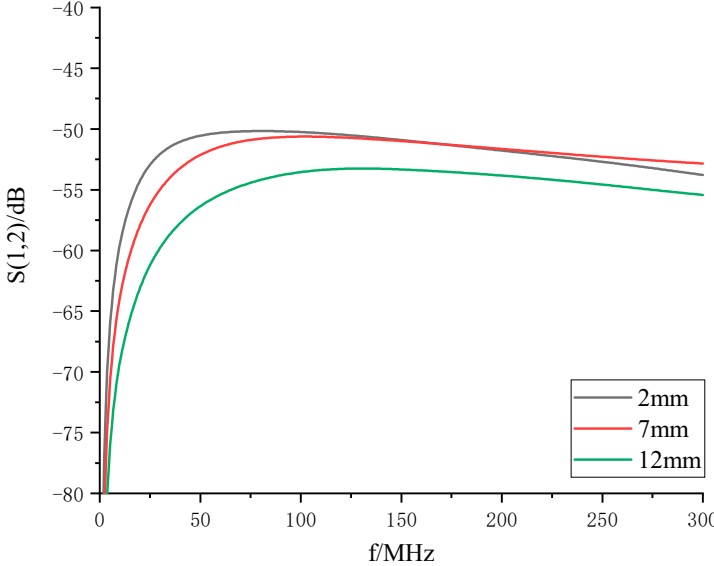

**Figure 8.** Frequency characteristics of probes with different line radius.

According to the design requirements of the probe, in the working band of the probe, the probe should have a smooth data output when receiving the magnetic field signal of the same field strength. It can be seen form Figure 8 that within 5 dB of frequency characteristic fluctuation, the lower cut-off frequencies of probe corresponding to different wire radius of 2 mm, 7 mm, and 12 mm are 18.00 MHz, 27.75 MHz, and 39.00 MHz, respectively. Therefore, when the probe loop radius and other settings are fixed, the lower cut-off frequency of the probe gradually decreases as the line radius decreases. Considering the designed lower cut-off frequency is 10 MHz, the line radius is set to 2 mm in the subsequent design.

### 3.3.2. Effect of Loop Radius on Frequency Characteristics

The probe model with a line radius of 2 mm was established, and the probe material properties, boundary conditions, and parts to be measured were set with reference to the previous subsection. Within the frequency range of 300 MHz, the magnetic field probes

with loop radius of 33 mm, 39 mm, and 45 mm were simulated, and the frequency response characteristics of the probes under the three loop radius sizes were shown in Figure 9.

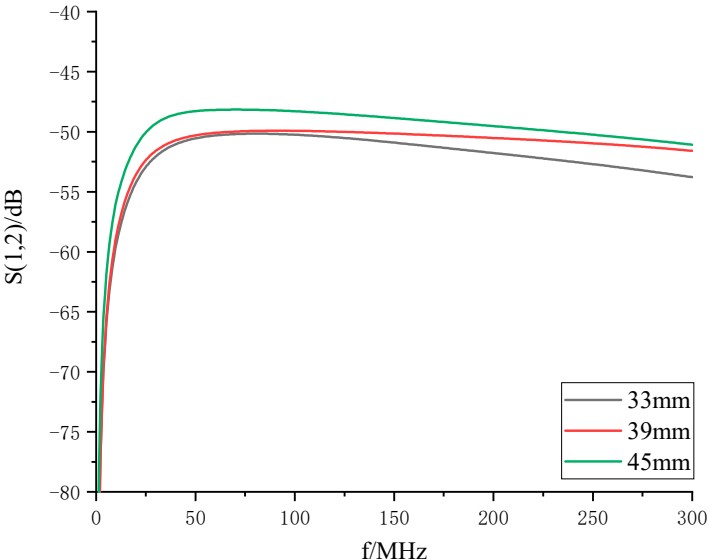

**Figure 9.** Frequency characteristics of the probe at different loop radius.

Within 5 dB of the fluctuation of frequency characteristics, the lower cut-off frequencies of the probes corresponding to different loop radius of 33 mm, 39 mm, and 45 mm were 18.00 MHz, 16.50 MHz, and 15.00 MHz, respectively, i.e., as the loop radius increases, the low-frequency characteristics of the probes become better and better, and their lower cut-off frequencies gradually decrease. However, if the probe size is too large, it will have a great impact on the electrical environment inside the switchgear. With reference to the above simulation results and the currently available dimensions of the switchgear built-in UHF PD sensors [33,34], the loop radius of the probe was set to 45 mm.

Combining with Equation (25), it can be obtained that when the loop radius is set to 45 mm, the frequency $f_{max}$ of the highest measurable signal of the probe is 212.31 MHz, which meets the required frequency range of 200 MHz in this paper.

### 3.3.3. Effect of Ferrite on Frequency Characteristics

Combined with Equation (22), it is known that the lower cut-off frequency can be reduced by increasing the inductance of the probe. In order not to continue to increase the size of the probe, a high permeability material can be used on the probe to collect the magnetic flux, considering the operating frequency band of the probe, the Ni-Zn ferrite material, which is widely used in the high frequency range. The structure of the probe is shown in Figure 10, and the permeability characteristic curve of the material used is shown in Figure 11.

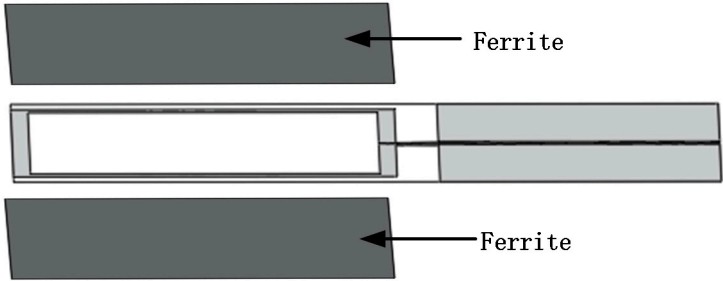

**Figure 10.** Probe structure with Ni-Zn ferrite sheets.

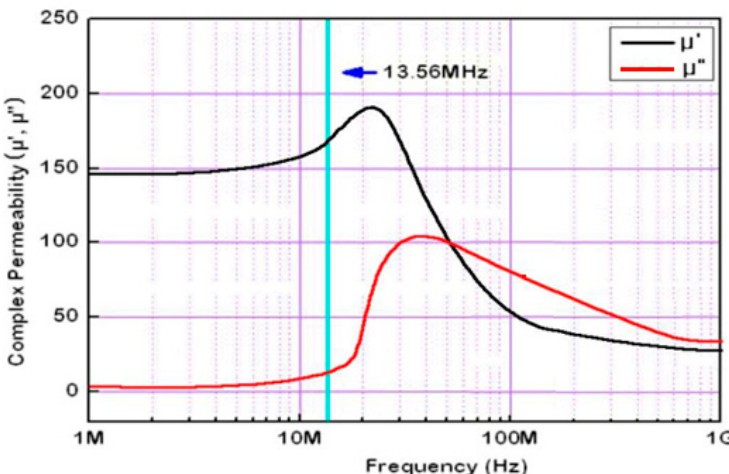

**Figure 11.** Magnetic permeability characteristic curves of Ni-Zn ferrite materials.

The probe model with a line radius of 2 mm and a loop radius of 45 mm was established, and the material properties of the probe, the boundary conditions, and the parts to be measured were also set up. The magnetic field probe with and without Ni-Zn ferrite sheet was simulated in the frequency range of 300 MHz, and the magnetic field intensity distribution of the probe with and without Ni-Zn ferrite sheet was obtained as shown in Figure 12, and the frequency response characteristics are shown in Figure 13.

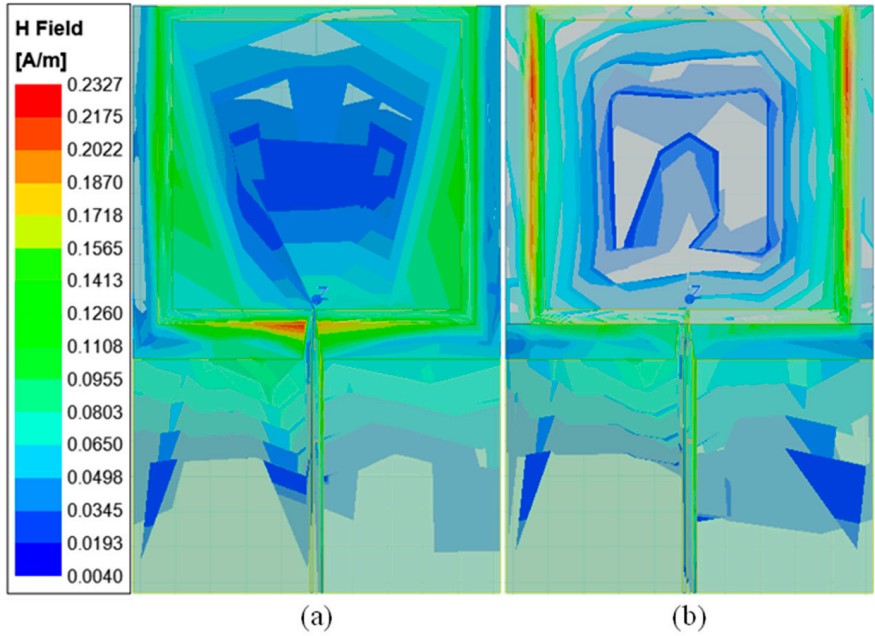

**Figure 12.** Magnetic field distribution at the top of the magnetic field probe with and without Ni-Zn ferrite sheet. (**a**) Without ferrite; (**b**) with ferrite.

From Figure 12, it can be seen that the maximum magnetic field intensity generated around the probe loop without and with the addition of Ni-Zn ferrite sheet was 0.2265 A/m and 0.2327 A/m, respectively, and the addition of Ni-Zn ferrite sheet increases the magnetic field intensity on the probe loop. Observing Figure 13, the lower cut-off frequencies of the probe with and without Ni-Zn ferrite sheet correspond to 9.75 MHz and 15.00 MHz, respectively, within 5 dB of the fluctuation of frequency characteristics. Therefore, the use of Ni-Zn ferrite sheet with high permeability can reduce the lower cut-off frequency point of the probe with the same probe structure size.

In addition, according to the design requirements, the larger the reflection coefficient of the probe input is required in the operating band, the probe port is still set to port 1 and $S_{11}$ is the transmission parameter from port 1 to port 1:

$$S_{11} = \left.\frac{b_1}{a_1}\right|_{a_2=0} \tag{27}$$

$$S_{11}(dB) = -20\log|\Gamma_L| \tag{28}$$

where $b_1$ represents the normalized outgoing wave voltage of port 1, $a_1$ represents the normalized incoming wave voltage of port 1, and $\Gamma_L$ is the reflection coefficient of port 1.

According to the design requirements, the larger the reflection coefficient at the probe input, the better, i.e., the closer $S_{11}$ is to 0 dB, the better. The probe model with line radius of 2 mm, loop radius of 45 mm, and Ni-Zn ferrite plate was established in HFSS simulation software. Within the frequency range of 300 MHz, $S_{11}$ of the probe is obtained as shown in Figure 14.

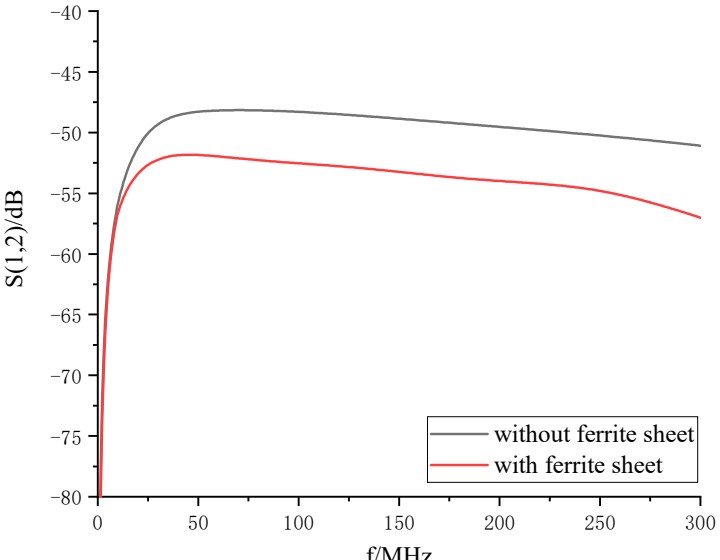

**Figure 13.** Probe frequency characteristics with and without Ni-Zn ferrite.

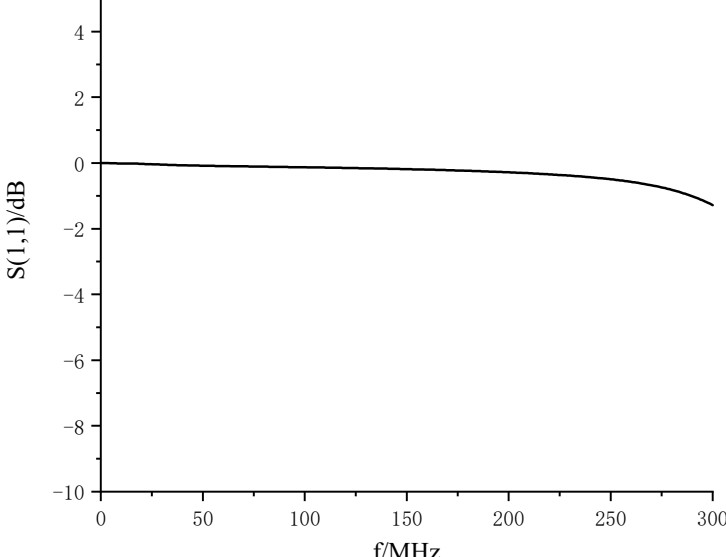

**Figure 14.** Transmission parameters of the probe port $S_{11}$.

It can be seen from Figure 14 that in the operating band, the return loss of the probe shows a good flat characteristic, and very close to 0 dB; that is, in the probe port, most of the signals are reflected, and it is difficult to have the signal radiated through the probe. It is also difficult to cause interference to the field to be measured.

## 4. Probe Performance Verification

According to the simulation analysis in the previous section, it is determined that the probe size that meets the design requirements is the line radius of 2 mm and the loop radius of 45 mm, and Ni-Zn ferrite sheets need to be pasted on the top and bottom surfaces of the probe. This section focuses on the actual measurement of the probe performance, as well as the verification of the probe performance on the simulated PD testbed.

### 4.1. Probe Performance Testing

The actual probe was fabricated as shown in Figure 15, and the dimensions of the probe were 166 mm long, 104 mm wide, and 2 mm thick. AV3656 vector network analyzer was used to measure the $S_{12}$ and $S_{11}$ parameters of this magnetic field probe. The test schematic is shown in Figure 16. The actual test results are shown in Figures 17 and 18.

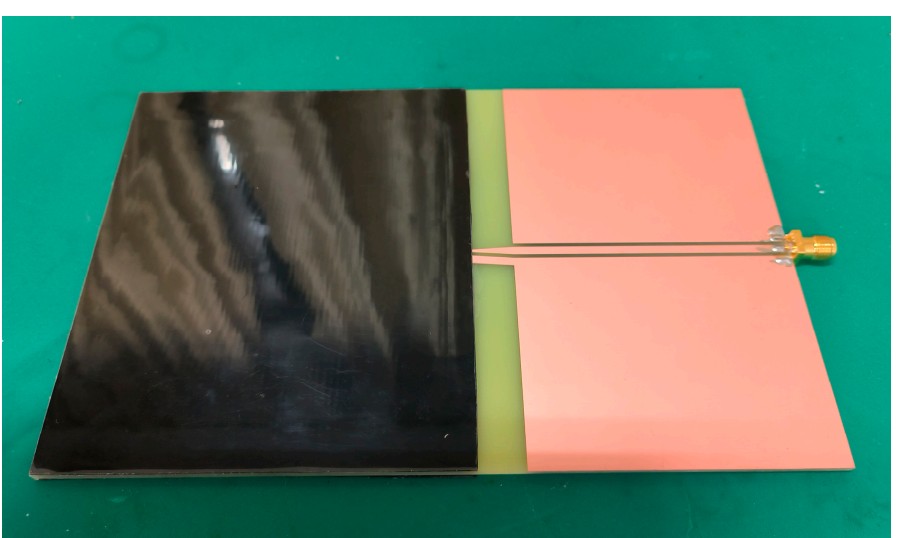

**Figure 15.** Physical picture of probe.

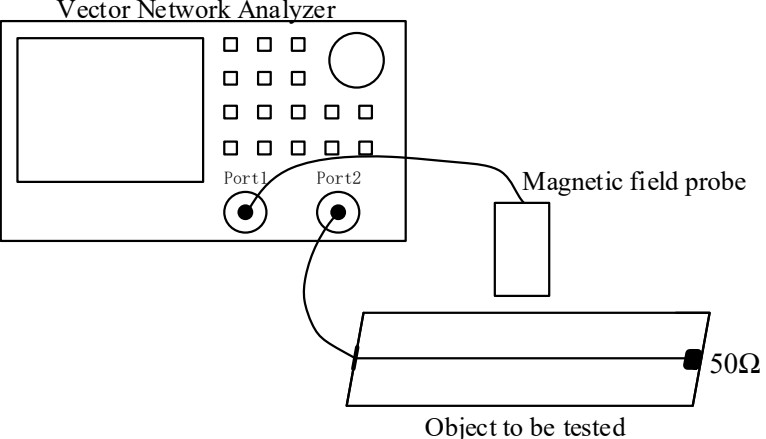

**Figure 16.** Schematic diagram of $S_{12}$ and $S_{11}$ parameters measured in practice.

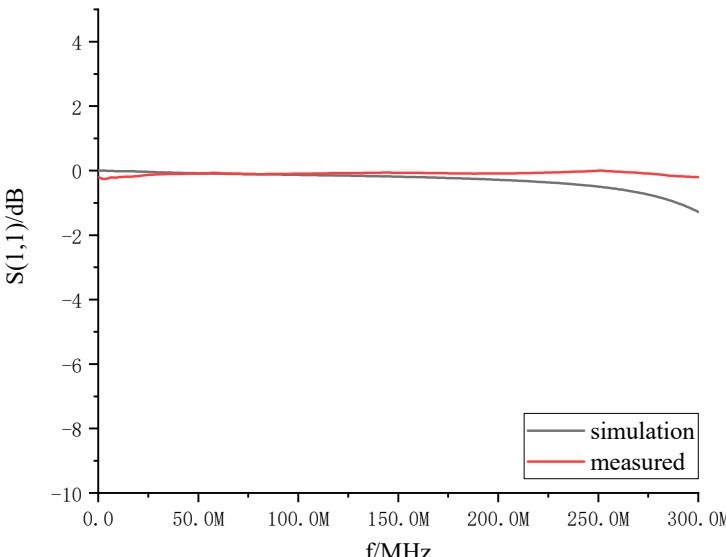

**Figure 17.** Comparison of test simulations of the $S_{11}$ parameters of the magnetic field probe.

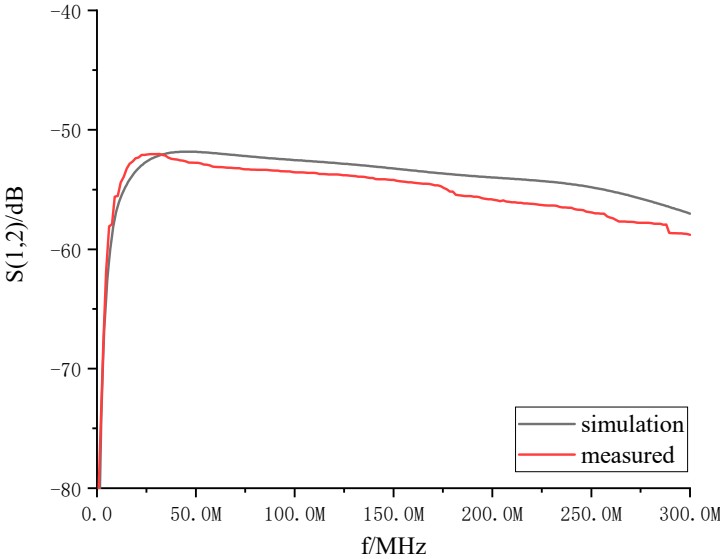

**Figure 18.** Comparison of test simulations of the $S_{12}$ parameters of the magnetic field probe.

Figure 17 shows the test and simulation comparison results of the $S_{11}$ parameters of the magnetic field probe. From Figure 17, it can be seen that the port reflection coefficient of the probe is very large, which makes its interference with the near field of the PD signal of the switchgear particularly small, and its test results are in good agreement with the simulation results.

Figure 18 shows the test and simulation comparison of the $S_{12}$ parameters of the magnetic field probe. As can be seen from Figure 18, the trends of the test and simulation results of the $S_{12}$ parameters are the same, with an error within 2 dB, but within the operating band of 10–200 MHz, the amplitude and frequency characteristics still fluctuate less than 5 dB, with good flatness.

Analysis of the causes of the error may have the following three points:

1.  The errors caused by the machining process of the actual probe. Due to the limitations of the processing machine, the processing accuracy is limited and errors are inevitably introduced during the process; in addition, the SubMiniature version A (SMA) connector needs to be connected during the actual test and the influence of the

SMA connector is not taken into account during the simulation process, which may also result in certain errors.

2.   Errors in the placement of the part to be tested and the probe in the actual measurement will also affect the actual results of the probe.

3.   The simulation software itself is subject to some errors, resulting in measured results that do not exactly match the simulation results.

### 4.2. Probe Detection PD Test

The probe detection PD test platform is built in the laboratory, and its wiring schematic diagram is shown in Figure 19. Among them, 1 is 220 V AC power supply, 2 is TDGC2-3KVA voltage regulator, 3 is GTB-5/50 dry-type test transformer, and 4 is grounding tip. The PD signal is collected by magnetic field probe 5 at a distance of 20 cm from the discharge source and obtained on oscilloscope 6. The oscilloscope model is Tektronix MDO3024. The time-domain and frequency-domain waveforms of the received PD signal are shown in Figure 20.

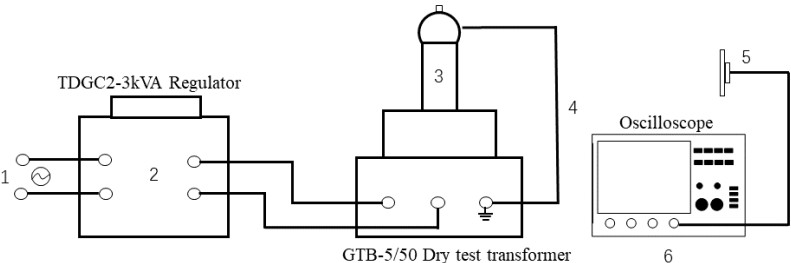

**Figure 19.** Schematic diagram of the probe detection partial discharge test platform.

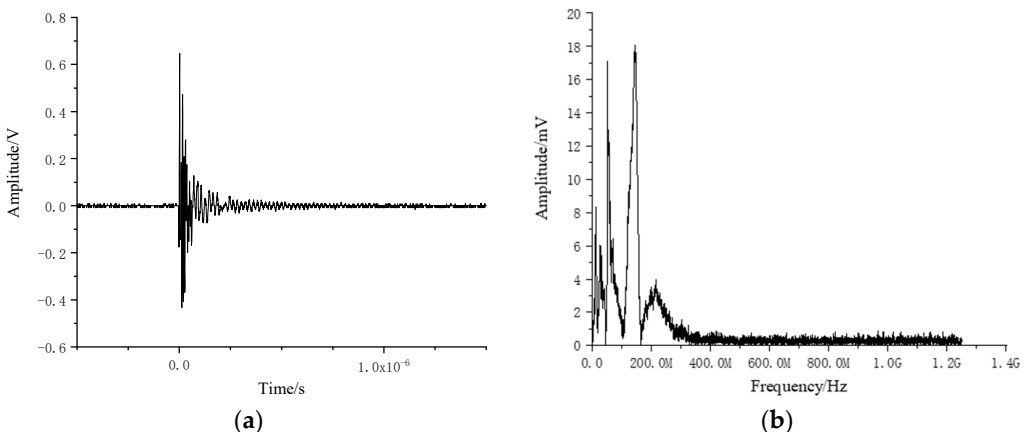

**Figure 20.** Magnetic field probe received PD signal. (**a**) time domain; (**b**) frequency domain.

As can be seen from Figure 20, the rising edge of the PD signal is very steep, with the peak value measured by the magnetic field probe approximating 0.65 V, followed by a gradual decay to background levels after approximately 0.25 μs. The results show that the signal received by the magnetic field probe has significant characteristics, which proves that the output signal of the magnetic field probe designed can accurately reflect the occurrence of PD. According to the spectrum analysis of the received signal, it can be seen from Figure 20 that the signal is mainly distributed below 200 MHz.

### 5. Conclusions

In this paper, based on the analysis of PD signal in the switchgear, a near-field magnetic field probe suitable for PD detection in switchgear is designed and a series of tests are carried out, with the following conclusions.

1. Based on the principle of PD pulse current generation, combined with the actual environment in the switchgear, the field characteristics are analyzed. and a new method of PD detection in the switchgear for near-field detection is proposed. The probe can be placed in the switchgear to realize online monitoring of PD and obtain the frequency band of PD signal energy concentration. It is expected to achieve quantitative and pattern recognition of PD in subsequent research.

2. The probe adopts the electric small loop structure, and its measuring principle and the factors affecting the amplitude frequency characteristics are analyzed in detail. On this basis, the HFSS software was used to simulate and discuss how to broaden the working frequency band of the probe on the basis of miniaturization of the probe. Finally, the structure and size of the probe meeting the design requirements were determined, and the performance of the probe was verified through actual testing. The results show that the designed magnetic field probe can accurately and significantly reflect the occurrence of PD.

3. The new method proposed in this paper can provide some reference for the follow-up research of PD near field detection. Later, the collected signals will be processed, that is, the noise separation of the probe and the quantitative study of the PD signal.

**Author Contributions:** Conceptualization, J.Z. and W.Z.; methodology, J.Z. and W.Z.; software, J.Z.; validation, J.Z. and Z.X.; formal analysis, J.Z.; investigation, J.Z.; resources, J.Z. and M.H.; data curation, J.Z.; writing—original draft preparation, J.Z.; writing—review and editing, J.Z.; visualization, J.Z. and Z.X.; supervision, C.S. and W.Z.; project administration, C.S., X.W. and W.Z.; funding acquisition, C.S. and W.Z. All authors have read and agreed to the published version of the manuscript.

**Funding:** "Research and development of new smart sensor technology to promote the development of green energy" (202104BN050011), and "Research and development of key technologies for conformal implantation of new intelligent power sensing for power grid main equipment" (YNKJXM20210075).

**Acknowledgments:** This research was supported by the "Research and development of new smart sensor technology to promote the development of green energy" (202104BN050011), and "Research and development of key technologies for conformal implantation of new intelligent power sensing for power grid main equipment" (YNKJXM20210075) funding.

**Conflicts of Interest:** The authors declare no conflict of interest.

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
