# Peer review of "Partial Discharge Detection Technology for Switchgear Based on Near-Field Detection"

_electronics, doi:10.3390/electronics12020336_

Round 1

Reviewer 1 Report

I find this paper interesting for the scientific community and I recommend it for publication in MDPI electronics. I have a minor suggestion to the authors. They should consider omitting some very basic results from the electromagnetic field theory and/or circuit theory. For example Eqs. (3)-(9) and (13) as well as (21)-(22) are part of a very basic undergraduate curriculum and seem a bit inappropriate in a research article. Although, I am teaching that stuff myself and like it general, maybe the authors could consider omitting some very basic stuff. 

Reviewer 2 Report

The description is not clear enough. The paper has to clarify the following questions.

1.     It looks like most of the literatures are in Chinese that not available. By the way, it might be better to indicate ‘In Chinese’ or something else, if the literature is not in English.

2.     Please show the meaning of acronym ‘PD’ the 1st time it is used.

3.     I am afraid the word ‘research ‘ in the title is redundant. By the way, do you mean technology of detecting partial discharge, or ‘technology of switchgear’? Please consider of it.

4.     Please give reference of HFSS.

5.     We have no idea what is what in this Fig. 1. Where does the PD occur?

6.     You might be able to omit some of the equations from (3) to (9). They are very basic for electromagnetism.

7.     Page 5, line 155, where is length ‘l’ in Fig. 5?

8.     The vectors from eq. (3) to (9) are without arrow, but the ones in (11) to (17) are with arrow? Please modify them.

9.     Page 7, line 231, the ‘A’ should be ‘S’.

10.  Where is the eq (33) come from?

11.  What is the probe radius for simulation in section 3.3.1?

12.  The arrows in Fig. 10 should be oppositely directed.

13.  Page 11, line 301, ‘will generated ‘  should be  ‘will be generated’

14.  Page 14, line 353 to 354, ‘… and the material properties of the probe, the boundary conditions….also set up’. Please show us the ‘set up’. By the, way, in line 355, ‘ the frequency range  of 300 MHz, …’, however, as shown in Fig. 11, the magnetic properties of the ferrite at 300 MHz is not so good, how do you deal with this problem?

15.   What is the object to be tested in Fig. 16?

16.  What is SMA in page 18?

17.  Where and how did the PD occur in Fig. 19? How far was it from the probe?

Reviewer 3 Report

This paper mainly discusses the challenges of designing a magnetic probe to detect a partial discharge (PD) inside of a switchingear in the near field. The obtained results are interesting; however the paper would benefit from improving the results sections.

The reviewer has the following remarks:

·         In the literature are three distinct types of defects caused by PD: internal, surface and corona defects. The authors should present which of the PDs phenomena can be detected by the proposed probe.

·         The authors described the magnetic probe’s line and loop radius influence over the S11 and S12 parameters, but there was no mentioning over the gap to Ground planes and the size of the ground area. Some details should be provided as in the end this information influences the final size of the magnetic probe.

·         Details of the measuring setups should be provided:

o   VNA proprieties

o   In the probe PD test it should be mentioned the distance between the source of the PD and the probe. How is the distance between the PD source and probe influence the measurements?

Reviewer 4 Report

1.       Avoid writing acronym like PD in the abstract. Why the word ‘Research’ is required in the title of the paper.

2.       Research objectives and gap are not clear. Authors should state the contributions of the research.

3.       Is Figure 1 own by the author? If not it requires reference. It also requires labeling.

4.       None of the modeling equations is provided with the reference. The authors must need to add the references.

5.       The probe frequency characteristics are compared only for ferrite and no-ferrite materials. However, there are other materials too. Can author add a detailed comparison?

6.       What are the major disadvantage of the proposed method/technology?

Round 2

Reviewer 2 Report

Thank you for revising the manuscript. 

Reviewer 3 Report

The authors have responded to the previous questions and inserted the requested details in the current manuscript.